# Effect of Ursolic and Oleanolic Acids on Lipid Membranes: Studies on MRSA and Models of Membranes

**DOI:** 10.3390/antibiotics10111381

**Published:** 2021-11-11

**Authors:** Sandrine Verstraeten, Lucy Catteau, Laila Boukricha, Joelle Quetin-Leclercq, Marie-Paule Mingeot-Leclercq

**Affiliations:** 1Université Catholique de Louvain, Louvain Drug Research Institute, Pharmacologie Cellulaire et Moléculaire, Avenue E. Mounier 73, UCL B1.73.05, 1200 Brussels, Belgium; sandrineverstraeten@gmail.com (S.V.); Lucy.Catteau@sciensano.be (L.C.); boukricha.laila@gmail.com (L.B.); 2Université Catholique de Louvain, de Duve Institute, Cellular Biology, Avenue Hippocrate 75, UCL B1.75.02, 1200 Brussels, Belgium; 3Université Catholique de Louvain, Louvain Drug Research Institute, Pharmacognosy, Avenue E. Mounier 73, UCL B1.73.05, 1200 Brussels, Belgium; joelle.leclercq@uclouvain.be

**Keywords:** MRSA, pentacyclic triterpenes, ursolic acid, oleanolic acid, ampicillin, synergy, models of membrane, biophysical properties, lipids

## Abstract

*Staphylococcus aureus* is an opportunistic pathogen and the major causative agent of life-threatening hospital- and community-acquired infections. A combination of antibiotics could be an opportunity to address the widespread emergence of antibiotic-resistant strains, including Methicillin-Resistant *S. aureus* (MRSA). We here investigated the potential synergy between ampicillin and plant-derived antibiotics (pentacyclic triterpenes, ursolic acid (UA) and oleanolic acid (OA)) towards MRSA (ATCC33591 and COL) and the mechanisms involved. We calculated the Fractional Inhibitory Concentration Index (FICI) and demonstrated synergy. We monitored fluorescence of Bodipy-TR-Cadaverin, propidium iodide and membrane potential-sensitive probe for determining the ability of UA and OA to bind to lipoteichoic acids (LTA), and to induce membrane permeabilization and depolarization, respectively. Both pentacyclic triterpenes were able to bind to LTA and to induce membrane permeabilization and depolarization in a dose-dependent fashion. These effects were not accompanied by significant changes in cellular concentration of pentacyclic triterpenes and/or ampicillin, suggesting an effect mediated through lipid membranes. We therefore focused on membranous effects induced by UA and OA, and we investigated on models of membranes, the role of specific lipids including phosphatidylglycerol and cardiolipin. The effect induced on membrane fluidity, permeability and ability to fuse were studied by determining changes in fluorescence anisotropy of DPH/generalized polarization of Laurdan, calcein release from liposomes, fluorescence dequenching of octadecyl-rhodamine B and liposome-size, respectively. Both UA and OA showed a dose-dependent effect with membrane rigidification, increase of membrane permeabilization and fusion. Except for the effect on membrane fluidity, the effect of UA was consistently higher compared with that obtained with OA, suggesting the role of methyl group position. All together the data demonstrated the potential role of compounds acting on lipid membranes for enhancing the activity of other antibiotics, like ampicillin and inducing synergy. Such combinations offer an opportunity to explore a larger antibiotic chemical space.

## 1. Introduction

In the past few decades, the antibiotic resistance of bacteria has emerged as a serious threat in human and veterinary medicine. As a critical example, the infectious diseases caused by multidrug-resistant bacteria, such as methicillin-resistant *Staphylococcus aureus* (MRSA), are becoming a very serious problem throughout the world (http://www.cdc.gov) (accessed on 26 June 2019). To tackle this situation, development of new antibiotics or therapeutic strategies against multi-drug resistant bacteria is urgently needed. One approach is to enhance the bacteria susceptibility to antibiotics. Receiving increasing attention, plant-derived chemicals have the potential to be used as therapeutics to enhance the activity of antibiotics against multidrug-resistant pathogens in addition or through their own effect(s) [1,2].

Pentacyclic triterpenes, a subfamily of the vast group of bioactive plant secondary metabolites, could be promising in this context. Two main representative compounds are ursolic acid (UA) (3β-hydroxy-urs-12-en-28-oic acid) and oleanolic acid (OA) (3β-hydroxy-olean-12-en-28-oic acid). Both show an antimicrobial activity against a large number of pathogens including carbapenem-resistant *Klebsiella pneumoniae* [3], colistin-resistant Enterobacteriaceae [4], vancomycin-resistant enterococci [5], *Porphyromonas gingivalis* [6], MRSA [7], or *Streptococcus mutans* biofilm [8]. In addition, stimulation of immunomodulatory properties [9], inhibition of peptidoglycan metabolism [10], prevention of cell division [10], and inhibition of efflux pumps [11] were reported. UA and/or OA also showed synergy with other antibiotics against a large panel of pathogens. This includes *S. aureus* and synergy with β-lactams [2], ampicillin and tetracycline [12], and norfloxacin [13]. On MRSA, synergy has been described with ampicillin and oxacillin [7]. Similar observations were also reported for *B. cereus* (synergy with ampicillin and tetracycline [12]), *Staphylococcus epidermidis* (synergy with β-lactams [2]), *Listeria monocytogenes* (synergy with β-lactams [2]), *Escherichia coli* (synergy with kanamycin [14]), and colistin-resistant *Enterobacteriacae* (synergy with colistin [4]).

The exact mechanism of action of pentacyclic triterpenes has not been fully elucidated. Hypothetically, Wang et al. [12] suggested an effect on proteins involved in the translation process, leading to the accumulation of mRNA and misfolded proteins with induction of ribonuclease and chaperon subunits. Furthermore, changes in the metabolism occur mostly to mediate the uptake and phosphorylation of carbohydrates and to control the metabolism in response to energy deficiency. Finally, UA could elicit the oxidative response in MRSA or alter the lipid cycle of cell wall biosynthesis by, e.g., interacting with undecaprenyl pyrophosphate [15,16]. These effects are likely subsequent to the alteration of membrane integrity [3,4,10,12,17]. As suggested by Prades [18], the effects of plant pentacyclic triterpenes on biophysical properties of membranes could play a role in their biological activity.

In this work, we aimed to first investigate the ability of UA and OA to bind to LTA, and to induce MRSA (ATCC33591 and COL) membrane permeabilization and depolarization. We also determined for UA, the most effective compound analyzed, its capacity to accumulate in *S. aureus* or to promote the passage of ampicillin through the membrane. Second, to explore the potential role of specific lipids in the effects induced by UA and OA, we used models of lipidic membranes (LUVs; Large Unilamellar Vesicles). On these liposomes, we characterized the effects of UA and OA on the biophysical membrane properties like fluidity, permeability and ability to fuse. Both on bacteria and on liposomes, we determined whether there was any difference in the effects induced by UA or OA.

## 2. Material and Methods

Ursolic acid (UA) and oleanolic acid (OA) (purity 98%) were obtained from Ava Chem (San Antonio, TX, USA) or Extrasynhese (Genay, France). 1-palmitoyl-2-oleoyl-*sn*-glycero-3-phosphoglycerol (POPG; Tm: −2 °C), 1,3-bis-(*sn*-3′-phosphatidyl)-*sn*-glycerol (cardiolipin; CL from *E. coli*) and 1,2-dioleoyl-*sn*-glycero-3-[phospho-rac-(3-lysyl(1-glycerol))] (chloride salt) (C18:1 lysyl-phosphatidylglycerol) were purchased from Avanti Polar Lipids (Alabaster, AL, USA). Octadecyl rhodamine B (R18) was purchased from Invitrogen (Paisley, Scotland, UK). Laurdan, calcein, DMSO and resazurin were ordered to Sigma, St. Louis, MO, USA. All solvents (analytical grade) were purchased from E. Merck AG. Tryptic soy agar (TSA) was bought to Difco (Difco, Richmond, CA, USA). High-performance thin-layer chromatography (HPTLC) plates precoated with silica gel 60 F254 were purchased from Merck KGaA (Darmstadt, Germany).

*Staphylococcus aureus* ATCC33591 (MRSA and β-lactamase producer; American Type Culture Collection, Manassas, VA, USA) and COL (HA-MRSA SCCmec type 1 strain COL) [19,20]; http://www.tigr.org/tdb/staphylococcus (accessed on 10 January 2021) were used.

### 2.1. Minimal Inhibitory Concentration (MIC) Determination

MICs were determined by a broth micro-dilution method according to the guidelines of the Clinical and Laboratory Standards Institute in cation-adjusted Mueller-Hinton broth (Ca-MHB) [CLSI, 2020].

Bacteria were cultured on tryptic soy agar (TSA), incubated overnight at 37 °C and adjusted to a bacterial density of 10^6^ CFU/mL as a starting inoculum. Triterpenic acids were solubilized in dimethylsulfoxide (DMSO) (10%) and then diluted in Ca-MHB to obtain a final concentration of 256 mg/L. Serial two-fold dilutions were made to obtain a concentration range from 0.25 to 128 mg/L. The plates were incubated during 20 h at 37 °C. To facilitate readings, bacterial growth was detected using resazurin, a blue phenoxazin dye that is reduced by viable bacteria in the pink fluorescent compound resorufin [21]. 30 µL of a 0.02% resazurin solution in Ca-MHB was added to each well. Plates were then incubated at 37 °C for 1 h in the dark. MIC corresponded to the lowest concentration of compounds for which the well color did not turn to pink. All tests were made in triplicate.

### 2.2. Fractional Inhibitory Concentration Indices (FICI) Determination

Fractional Inhibitory Concentration Indices (FICI) were determined by the checkerboard method in Ca-MHB [22]. In a 96-well plate, the β-lactam antibiotic (ampicillin) was serially diluted starting from a final concentration of 2 × MIC along the ordinate. The triterpenic acid was serially diluted in Ca-MHB along the abscissa using 2 × MIC as the highest final concentration. The bacterial suspension (final inoculum 0.5–1 × 10^6^ CFU/mL) was added to the wells. After 20 h of incubation at 37 °C, 30 µL of resazurin solution (0.02% resazurin) were added to the wells for 40 min. Bacterial growth was quantified by recording fluorescence at 590 nm [23]. The lowest concentration of compound that resulted in inhibition of bacterial growth was used to determine the MICs.

Interactions between antibiotics were then evaluated using the FIC Indices, calculated as the sum of the Fractional Inhibitory Concentrations (FICs) as follows: FICI = FIC A + FIC C, where FIC A is MIC of the antibiotic in the combination/MIC of the antibiotic alone and FIC C is MIC of the compound in the combination/MIC of the compound alone [22]. The combination was considered as synergistic for FICI ≤ 0.5, additive for 0.5 < FICI ≤ 1, indifferent for 1 < FICI ≤ 4 and antagonistic for FICI > 4 according to the European Committee for Antimicrobial Susceptibility Testing (EUCAST). All tests were made in triplicate.

### 2.3. BODIPY™-TR-Cadaverine Displacement Assay

Binding affinity to lipoteichoic acid (LTA) from *S. aureus* was investigated in a cell free system as well as on bacteria, by using BODIPY™-TR-cadaverine displacement assay. Quenching of fluorescence intensity was observed when the probe was bound to LTA. Displacement of probe into the solution leads to enhancement of its fluorescence. Assays were performed as described in Swain et al. [24]

### 2.4. Ursolic Acid and Ampicillin Uptake in S. aureus ATCC 33591 and COL

*S. aureus* ATCC33591 and COL were cultured on tryptic soy agar (TSA) and colonies were incubated overnight at 37 °C in Ca-MHB upon shaking (130 rpm). The bacterial density was adjusted as 10^9^ CFU/mL. Bacteria were incubated at 37 °C with UA (40 µg/mL as final concentration), ampicillin (32 µg/mL; 0.5 MIC), or both for increasing times (30, 60 and 120 min). In some experiments, bacteria were preincubated with UA (30 min) and thereafter with ampicillin (30 min). The opposite protocol was also tested. At the end of incubation, the tubes were centrifuged at 4000 rpm for 7 min at 4 °C. The pellets were resuspended in 5 mL of cold sodium phosphate buffer (pH 7.0) and centrifuged again at 4000 rpm for 7 min at 4 °C. After twofold, the pellets were resuspended in 300 µL of water MilliQ. Two hundred µL were used for preparing cell lysates (sonication) and the remaining 100 µL were kept for CFU determination.

The concentration of UA from lysates was estimated by HPTLC and densitometric analysis, after liquid/liquid extraction with ethyl acetate [25,26]. Fifty µL of sonicated bacteria (lysates) were added to 1 mL of ethylacetate. The mixture was vortexed for 2 min, centrifuged at 2000× *g* for 10 min at 4 °C. The upper phase was fully recovered and evaporated. The residues were resuspended in chloroform:methanol, vortexed, centrifuged 12,000× *g*, 4 °C for 5 min.

The amount of UA was estimated by the modified method of Sethiya and Mishra [27] validated on the parameters such as linearity (1–1000 ng/µL), limit of detection (2–3 ng/µL) and quantification (10 µg/mL), specificity, precision, accuracy, recovery, and robustness by Wojciak–Kosior [28].

The HPTLC plates (20 cm × 10 cm pre-coated HPTLC silica gel 60 F254) were cleaned by predevelopment by using methanol and dried on hotplate at 120 °C for 20 min before loading the samples. The deposit was performed with a 10 µL Hamilton syringe. Two μL of standard solutions in methanol and samples of extracts were spotted as 3 mm-wide bands (track distance: 10 mm, distance from the left edge: 15 mm). After loading, HPTLC plates were pre-derivatized with 1% iodine solution in chloroform up to a distance of 1.5 cm and were placed in the dark for 5 min. The plates were dried on a hot plate at 120 °C for 5 min and in the oven at 60 °C for 10 min to remove the excess of iodine. Then the samples were developed in a saturated CAMAG twin trough glass chamber in the mobile phase (n-hexane: ethyl acetate: acetone 8.2:3.6:0.2 (*v/v*) mixture) up to a distance of 7 cm. After drying, the plates were sprayed with 10% (*v/v*) H_2_SO_4_ in ethanol, dried in room atmosphere for 10 min by shaking and were then heated for 3 min at the temperature of 120 °C. UA gave well-resolved spots at Rf 0.62. The plates were scanned by CAMAG-TLC scanner-3 (Mettler-Toledo, Zaventem, Belgium) within 30 min; afterwards a progressive degradation was observed. Quantitative evaluation of the plate (WinCATS software (version: 1.3.0)) was performed by keeping slit dimensions to 4 × 0.33 mm, scanning speed, 20 mm s^−1^ at a wavelength of λ = 535 nm. The peak areas were recorded and calibration curves were obtained by plotting peak area versus concentrations of the standards (15, 20, 30, 40, 50, 60 ng/µL).

The amount of ampicillin was measured by a fluorometric assay as described by Jusko [29]. The assay was linear for ampicillin concentrations ranging from 0 to 0.8 µg/mL.

250 µL TCA (trichloroacetic acid 40%) was added to 200 µL cellular samples and the mixture was centrifuged to precipitate the proteins. The supernatant was recovered and 100 µL formaldehyde (formaldehyde 7%: from 37% formaldehyde diluted in citric acid buffer pH = 2; 0.5 mM) was added. The mixture was placed in a water bath at 90 °C for 2 h, cooled down before addition of 200 µL NaOH 2N and fluorescence measurement (λ_exc_ 346 nm; λ_em_ 422 nm).

### 2.5. S. aureus Membrane Permeabilization as Fluorescence of Propidium Iodide

The permeabilization of the bacterial membrane was determined with a membrane-impermeable fluorescent dye (Propidium Iodide; PI), which can enter permeabilized bacteria only [30]. A stock solution of PI (3 mM in water) was diluted 10^3^-fold with the bacterial suspension (A_620_ = 0.05). UA and OA in HEPES buffer 5 mM, pH 7.4, at final concentrations ranging from 0 to 100 µg/mL, were added to the PI-containing bacterial suspension in 96-well microplates. The fluorescence intensity was measured with a SpectraMax M3 microplate reader at 25 °C after 15 min of stabilization (Molecular Devices, Sunnyvale, CA, USA) for excitation and emission wavelengths of 520 and 627 nm, respectively. The data were normalized based on the fluorescence intensity measured in the presence of alexidine (5 µM) (positive control, 100%).

### 2.6. S. aureus Membrane Depolarization as Fluorescence of DiSC3(5)

The depolarization of bacterial membranes was investigated by using the membrane potential-sensitive fluorescent probe DiSC3(5) [31,32]. *Staphylococcus aureus* (MRSA and COL) were grown and isolated as described previously. The bacterial suspension (A_620_ = 0.3) was washed twice first with HEPES buffer (5 mM pH 7.4, enriched with glucose 5 mM) and second with HEPES buffer (5 mM, pH 7.4) enriched with 5 mM glucose and 100 mM KCl. The DiSC3(5) probe was added (final concentration 800 nM) to the bacterial suspension (DO_620_ = 0.05) and the mix was incubated in the dark for 30 min at 37 °C. UA and OA were distributed in a 96-well microplate; thereafter, the bacterial suspension containing DiSC3(5) was added to obtain a final derivative concentration ranging from 0 to 100 µg/mL. The K-specific ionophore, valinomycin, was used as a positive control (10 µM) [33]. Readings were performed with excitation and emission wavelengths of 630 and 680 nm, respectively. A preliminary experiment, without bacterial cells, was performed to ensure that the presence of the pentacyclic triterpenes had no effect on DiSC3(5) fluorescence.

### 2.7. Large Unilamellar Vesicles (LUVs) Preparation

LUVs were prepared with the extrusion method from multilamellar vesicles. Phospholipids (5 mg/mL in CHCl_3_/CH_3_OH (2:1, *v*/*v*)) were mixed in the desired molar ratio, namely POPG/CL, 85:15; POPG/CL, 60:40. Multilamellar vesicles were prepared according to the freeze thawing method in an aqueous buffer (10 mM Tris-HCl, 159 mM NaCl, pH 7.4), eventually containing calcein (73 mM) or KCl (150 mM). LUVs were then obtained with 10 successive extrusions of multilamellar vesicles through two superimposed track-etch polycarbonate membranes (pore size 100 nm; Whatman Nucleopore, Corning Costar Corp., Badhoevedorp, The Netherlands) using a 10 mL Thermobarrel^R^ extruder (Lipex Biomembranes, Vancouver, BC, Canada). For calcein-filled LUVs, Sephadex^R^ gel filtration was used to remove the unencapsulated calcein with the minicolumn centrifugation technique [34,35]. The phospholipid concentrations in LUVs suspensions were determined with a Bartlett phosphate assay [36]. Iso-osmotic buffers were used to make any necessary adjustments.

### 2.8. Membrane Fluidity as DPH Anisotropy Measurements

The influence of UA and OA on the fluidity of the hydrocarbon domain of the bilayer was followed by monitoring the steady-state fluorescence polarization of DPH probes [37]. DPH is thought to reside in the hydrophobic core of the membrane [37,38,39,40]. DPH was dissolved to a final concentration of 100 μM in tetrahydrofuran and incorporated to the sample with the lipids before evaporation to a molar ratio 300:1 (lipid: DPH). The total phospholipid concentration of each preparation was adjusted to a final value of 5 μM with 10 mM Tris–HCl, 159 mM NaCl, pH 7.4. UA and OA were incubated for 5 min at 37 °C with liposomes in the dark. Anisotropy (r) of samples was determined upon time (60 min). All fluorescence determinations were performed on a LS 55 fluorescence spectrophotometer using λ_exc_ and λ_em_ of 381 nm and 426 nm, respectively and slits fixed at 2.5 and 11 nm, respectively; r values were determined as shown in the following equation:r = I_VV_ G⋅I_VH/_I_VV_ + 2⋅G⋅I_VH_

where I_VV_ is the fluorescence intensity when angle between polarizers is 0°, I_VH_ is the fluorescence intensity when angle between polarizers is 90°, and G is an inherent factor to the fluorometer used.

### 2.9. Lipid Phases as Laurdan Generalized Polarization Studies

The effect of UA and OA on the gel–liquid crystalline phases of the phospholipids at the level of glycerol backbone was determined by monitoring the Laurdan excitation generalized polarization. Laurdan is a polarity-sensitive probe [41], located at the glycerol backbone of the bilayer with the lauric acid tail anchored in the phospholipid acyl chain region [42]. The bilayer fluidity-dependent fluorescence spectral shift of Laurdan due to dipolar relaxation phenomena was monitored. Upon excitation, the dipole moment of Laurdan increases noticeably and water molecules in the vicinity of the probe reorient around this new dipole. When the membrane is in a fluid phase, the reorientation rate is faster than the emission process and, consequently, a red-shift is observed in the emission spectrum of Laurdan. When the bilayer packing increases, part of the water molecules is excluded from the bilayer and the dipolar relaxation of the remaining water molecules is slower, leading to a fluorescent spectrum, which is significantly less shifted to the red [43]. Fluorescence determinations were carried out using a thermostated Perkin-Elmer LS55 fluorescence spectrophotometer at an excitation wavelength of 340 nm. The lipid concentration of liposomes was adjusted to 5 μM with 10 mM Tris–HCl, 159 mM NaCl, pH 7.4 and Laurdan was added from a 5 × 10^−3^ M stock solution of DMF to give a lipid:probe ratio of 300 (mol:mol). Drugs were added to liposomes to obtain final concentrations of 0.4, 0.8, 1.6 µg/mL and incubated under continuous agitation at 37 °C out of light for 60 min. Generalized polarization (GP) from emission spectra was calculated using equation below:GPex = I_440_ − I_490_/I_440_ + I_490_

where I_440_ and I_490_ are the fluorescence intensities at emission wavelengths of 440 nm (gel phase) and 490 nm (liquid crystalline phase), respectively, at a fixed excitation wavelength of 340 nm.

### 2.10. Membrane Permeabilisation as Calcein Release Measurements

Membrane permeabilization was followed by monitoring the leakage of entrapped, self-quenched calcein from liposomes upon measuring the increase of fluorescence signal subsequent to its dilution in the external medium [44]. The liposome total phospholipid concentration was adjusted to a final concentration of 5 μM with iso-osmotic buffer (10 mM Tris–HCl, 159 mM NaCl, pH 7.4). Liposomes were exposed to UA and OA at 37 °C at the desired final concentration (0.4, 0.8, 1.6, 6.4 and 12.8 µg/mL) and for the suitable time with continuous stirring and protection from light. All fluorescence determinations were performed on a LS 55 fluorescence spectrophotometer (Perkin–Elmer Ltd., Beaconsfield, UK) using λ_exc_ and λ_em_ of 476 nm and 512 nm, respectively, and slits fixed at 3 nm. The percentage of calcein released under the influence of drug was defined as:(F_t_ − F_contr_)/(F_tot_ − F_contr_) × 100
where F_t_ is the fluorescence signal measured at time t in the presence of drug, F_contr_ is the fluorescence signal measured at the same time in the absence of drug, and F_tot_ is the total fluorescence signal obtained after complete disruption of liposomes by Triton X-100 at a final concentration of 2% (checked by quasi-elastic light spectroscopy).

### 2.11. Membrane Fusion as R18 Dequenching Measurements

R18 (Octadecyl Rhodamine B), a lipid-soluble probe, was incorporated into a lipid membrane as its fluorescence became self-quenched in direct proportion with its concentration (in a 1–9-mol % range). A decrease in its surface density [45] is associated with an increase in the fluorescence intensity of the preparation. Labeled liposomes were obtained by incorporating R18 in the dry lipid film at a molar ratio of 5.7% with respect to the total lipids and diluted to a concentration of 5 µM in average lipid. They were then mixed with unlabeled liposomes and adjusted to the same concentration, at a ratio of 1:4. The variation of the fluorescence intensity of the preparation was thereafter followed at room temperature for 5 min, using a ƛ_exc_560 nm and ƛ_em_583 nm.

### 2.12. Liposome Size as Dynamic Light Scattering Measurements

Dynamic light scattering measurements were used to control the size and polydispersity of the LUVs. The apparent average diameter of the suspended particles was monitored by using a Malvern Zetasizer Nano ZS (Malvern Instruments, Ltd., Worcestershire, UK) equipped with a helium-neon laser and added back scattering detection at 173°. After the addition of a small aliquot (10 µL) of UA and OA at increasing concentrations (from 0.4 to 12.8 µg/mL) to LUVs (1 mL, 5 mM lipids), the mixture was diluted ten times in Tris HCl/NaCl buffer. After 5 min, dynamic light scattering measurements in a polystyrene cuvette were performed. All measurements were repeated at least three times. Size distribution was obtained by accumulating three measurements consisting of 13 successive runs of 10 sec. The Zetasizer Nano Software (supplied with the apparatus) was used to analyze the normalized intensity autocorrelation functions with the CONTIN algorithm.

### 2.13. Statistical Analysis

Data are expressed as means ± SEM. All statistical analyses were performed under GraphPad Prism version 4.3 for Windows (GraphPad Software, San Diego, CA, USA). If each condition is compared to the control, one-way ANOVA with Dunnett’s post-test was used. In case of multiple comparisons, a one-way ANOVA with Bonferroni’s multiple comparison post-test was selected.

## 3. Results

### 3.1. MIC and Synergistic Effect of Ursolic or Oleanolic Acids with Ampicillin

Both triterpenic acids, UA and OA, were active against MRSA (ATCC33591 and COL) with MICs ranging from 16–8 mg/L for UA and 32–16 mg/L for OA, respectively. These values are lower than the MIC of ampicillin against MRSA (ATCC33591 and COL) (64 and 32 mg/L) (Table 1).

Fractional Inhibitory Concentration Indices (FICI) (Table 1) were 0.38–1.00 for AMP-UA (ATCC and COL strains), 0.31–1 for AMP-OA (ATCC strain) and 0.25–1.00 for AMP-OA (COL strain), suggesting synergy or additivity with a decrease of ampicillin MIC in the presence of UA or OA.

To gain an insight into the molecular mechanisms involved in the effects induced by UA and OA on MRSA membranes, we explored the ability of these two pentacyclic triterpenes to bind to lipoteichoic acids (LTA), and to induce MRSA membrane permeabilization and depolarization.

### 3.2. Ursolic and Oleanolic Acids Bound to Lipoteichoic Acid (LTA) of MRSA

On MRSA (ATCC33591 (Figure 1 left), COL (Figure 1 right)), the evolution of the fluorescence of BODIPY™-TR-cadaverine induced by UA and OA, was monitored in comparison with the effect induced by alexidine (positive control) and neamine (negative control). For both pentacyclic triterpenes, an effect was already observed at the lowest concentration tested (25 µg/mL). The effect was dose-dependent with a larger effect induced by UA as compared with OA (Figure 1). In the presence of alexidine, the effect is higher in COL strains as compared to ATCC33591 strains (Figure 1). This difference was not observed with the pentacyclic triterpenes (Figure 1).

### 3.3. Ursolic and Oleanolic Acids Induced MRSA Membrane Permeabilization and Depolarization

To explore membrane permeability, fluorescence of propidium iodide (PI) was monitored with alexidin (5 µM) which was used as a positive control (100%) (Figure 2A,B). Both UA and OA induced a dose-dependent permeabilization with a plateau value reached around 20 µg/mL. No difference was observed for MRSA ATCC 33591 (Figure 2A) and COL (Figure 2B) strains. On both strains, the effect induced by UA was largely higher as compared to that induced by OA (50% versus ≤10% at the plateau value).

Fluorescence of the membrane potential-sensitive probe DiSC3(5) was followed with valinomycin (10 µM) used as a positive control. Results are expressed in percentage of the effect induced by this ionophore (Figure 2C,D). Globally speaking, concentrations required to induce membrane depolarization were lower as compared to those needed for membrane permeabilization. The maximal percentage of membrane depolarization induced by UA and OA were close for both ATCC33591 and COL strains. On ATCC33591, UA was more efficient than OA and around 20% of membrane depolarization was already observed at 0.5 µg/mL of UA.

### 3.4. Ursolic Acid Did Not Accumulate in S. aureus and Did Not Increase the Accumulation of Ampicillin

Cellular accumulation in *S. aureus* ATCC 33591 of UA or ampicillin, alone or in combination, was measured after 30 min of incubation. Figure 3A illustrates the accumulation of UA and the potential effect of ampicillin on this accumulation and Figure 3B, the accumulation of ampicillin and the potential effect of UA. Results from co- and pre-incubation were compared.

Whatever the protocol used (co-incubation or pre-incubation), we did not observe any significant effect of ampicillin on UA accumulation (Figure 3A) nor of UA on ampicillin accumulation (Figure 3B).

With the further aim to acquire molecular understanding of these cellular effects, we explored the capacity of UA and OA to alter biophysical membrane properties including membrane fluidity, membrane permeability, and lipid membrane fusion. We used liposomes (LUVs) as models of membranes. LUVs were prepared from phosphatidylglycerol (POPG) and cardiolipin (POPG:CL 85:15 or 60:40), two major lipids found in Gram-positive bacteria [46]. We focused on cardiolipin, a dimeric lipid, which can organize in domains at the septum of bacteria during cytokinesis and which plays a role in PBP2 location. We also questioned the role of lysylphosphatidylglycerol on membrane permeabilization.

### 3.5. Ursolic and Oleanolic Acids Decreased the Fluidity of Lipid Membrane Models

DPH fuorescence anisotropy (r) (Figure 4A,B) and the Laurdan Generalized Polarization (Gpex) (Figure 4C,D) upon incubation with increasing concentrations of UA and OA were followed.

Non-dependent upon the ratio between POPG and cardiolipin (85:15 or 60:40), we showed that both UA and OA increased the fluorescence anisotropy (Figure 4A,B) and generalized polarization (Figure 4C,D). No significant difference was observed when comparing the effect induced by UA or OA.

### 3.6. Ursolic and Oleanolic Acids Increased Permeability, Fusion and Size of LUVs

Results monitoring the release of calcein and the relief of fluorescence self-quenching of R18-labeled liposomes, induced by UA and OA on LUVs composed of POPG:CL (85:15 or 60:40) showed that, regardless of lipid composition, and for both the membrane permeabilization (Figure 5A,B) and ability of membranes to fuse (Figure 5C,D), the effect was dose-dependent. UA showed a higher effect as compared with OA. Interestingly, the maximum percentage of calcein release and fluorescence dequenching were around 25% and 40%. No major differences are observed upon changes in POPG:cardiolipin ratios. Similarly, no change was observed when lysylphosphatidylglycerol, a lipid related to daptomycin-resistant MRSA [47], was included in the LUV composition as illustrated for the calcein release assay (Appendix A).

At the lowest concentrations of UA and OA used, the average diameter of LUVs (POPG:CL (85:15; 60:40)) was around 100 nm. Increasing the concentrations, two populations appeared, one centered around 100 nm and the other around 1000 nm (Figure 6). This population size profile suggests a fusion process as compared to aggregation in which the heterogeneity of the size population is higher [48,49]. The percentage of the liposome population with larger size increased with the increase of UA/OA concentrations. This behavior was observed for both pentacyclic triterpenes. The effect was slightly more important for UA as compared with OA and liposomes composed from POPG:CL 60:40 were more prone to fuse in comparison with POPG:CL (85:15).

## 4. Discussion

Pentacyclic triterpenoids such as UA (ursolic acid) and OA (oleanolic acid) and their derivatives were discovered earlier as potential alternatives to antibiotics against a broad spectrum of pathogens [9,12,15,50]. Additionally, UA and OA showed synergy with β-lactams against MRSA [7], which could be critical for facing the loss of antibiotic activity. After having confirmed the interest of UA and OA, in combination with ampicillin, against MRSA, we elaborated on the potential mechanism involved and characterized the interaction between UA and OA with lipoteichoic acids of *S. aureus* and the consequences of this interaction for biophysical membrane properties.

In MRSA, mechanisms involved in synergy are diverse, including a decrease in β-lactamase activity. To investigate the potential role of β-lactamase in combination of PBP2a, we used two different strains. They differ in their expression of resistance mechanisms, ATCC33591 expresses PBP2a and β-lactamase whereas COL only expresses PBP2a [19]. On both strains (ATCC 33591 and COL), we demonstrated the activity of UA and OA, with MICs ranging from 8–16 mg/L for UA to 16–32 mg/L for OA. These results indicated a strong antibacterial activity against MRSA in agreement with data from other researchers as well [5,51,52]. A partial synergistic effect between UA (or OA) and ampicillin was observed with an FIC Index for ampicillin-UA/OA of 0.38/0.31–1 and 0.25/0.31–1 on ATCC and COL strains, respectively. The slightly higher effect on the COL strain suggests either a major role of PBP2a in the activity of pentacyclic triterpenes or a more general effect, like a membranous effect. The latter could involve general modifications of biophysical membrane properties or an effect on peculiar lipids like cardiolipin, required for proper activity of PBP2a or described for playing a role to circumvent antibiotic (daptomycin) and immune attacks in *S. aureus* [53].

As an attempt to elucidate the mechanism of action of UA and OA, we characterized on MRSA the capacity of UA and OA to interact with lipoteichoic acids, and to induce bacterial membrane permeabilization and depolarization. To our knowledge, this is the first time these three events were investigated by keeping the same experimental model. We confirmed the ability of UA and OA to induce membrane permeabilization [3,4,54] and membrane depolarization [3]. Interestingly, we demonstrated that the concentrations required to induce depolarization is lower than that needed to result in membrane permeabilization, while both are lower than what is required to reach the plateau value in BodipyTM-TR-cadaverin displacement as a reflection of the interaction with LTA. Interestingly, membrane permeabilization induced by UA did not result in its own accumulation in bacteria, nor that of ampicillin.

To explore the potential role of peculiar lipids, we demonstrated on liposomes with varying ratio of POPG and cardiolipin, an increase of membrane rigidity, membrane permeabilization and ability of membranes to fuse. Regarding the molecular mechanisms, data suggest the following conclusions.

First, the effects induced by UA/OA could not be primarily driven by the membrane charge. Indeed, neither the changes in POPG:cardiolipin ratios (85:15 vs. 60:40) nor the addition of lysylphosphatidylglycerol (Appendix A) affect the interaction of UA and OA with LTA or the release of calcein.

Second, UA and OA induced a decrease of membrane fluidity. This is in agreement with experiments performed by Han and coll. who aimed to investigate the effect of UA on crystalline and liquid-crystalline phase as well [17]. Even our experiences do not allow us to know whether the stiffening is an insertion-based effect (like cholesterol) or a clustering effect, Lorincz and coll. demonstrated by using FTIR and DPPC, a broadening gel-to-liquid crystalline phase with the phase transition temperature shifted towards higher values and an increase of gauche isomers of acyl chains above Tm upon the presence of UA. The mechanism is likely different to that of cholesterol since cholesterol, in contrast with UA, abolished completely the main transition and the hidden formation of gauche conformers [55]. Differences in the effect induced by UA and OA could result in differences in the propensity of UA and OA to induce interdigitated lipid structure at low temperatures, as observed for membrane-active peptides. The hydrocarbon chains would be shielded from water by UA and OA in the gel phases of negatively charged phosphatidylglycerol and cardiolipin. It has been suggested that, although this structure was observed only in the gel phase, its effects may propagate into the physiologically more relevant fluid phase in terms of membrane thinning and membrane perturbation. Regarding the clustering effect, disintegration by UA of membranes enriched in cardiolipin was reported in the literature [56,57,58]. As we used a mixture of POPG and cardiolipin from *E. coli* (and lysylphosphatidylglycerol), significantly different thermotropic behaviors with multiple transitions are expected when mixed. The mixing/demixing behavior must be driven by a variety of different interactions occurring between the headgroups of the lipids.

Thirdly, UA showed a higher effect as compared with OA, whatever the model used (bacteria or model of membranes) and the effect investigated. This is in accordance with the higher antibacterial activity of UA and with data in the literature [59]. Both ursane and oleanane skeletons are built of five cyclohexane rings, which are fused in the *trans* conformation with the exception of C18 carbon atom where the conformation is *cis*. The double bond between C12 and C13 is typical for the terpenes belonging to these groups. The *cis* junction results in a non-planarity of UA and OA (as opposed to compounds with a lupeol skeleton). Their configuration originated from the van der Waals type interaction between the ester group of the lipid and the –OH group of UA and OA [55]. Ursanes and oleananes differ only in the location of one methyl group [58] and thus in number and location of chiral centers [60]. This could result in differences in the interaction with lipids and the possibility of H-bond formation with the cyclopropane ring of the hydrocarbon chain of phospholipids [61]. The degree of permeation of each molecule into the bilayer should be a compromise between the number of hydrophobic contacts and the cost of losing some H-bonds in water. In turn, this can result in differences in the ability to interact with LTA (Appendix A) and to alter biophysical membrane properties. Further studies, including with ursolic derivatives [62], could be conducted to explore in more detail structure–activity relationships [60,63,64].

Reduction in membrane hydration as revealed by an increase in generalized polarization of Laurdan induced by UA and OA, also raised the question regarding a potential effect on lipid polymorphism, especially a decrease in stalk formation and thus in fusion [65]. This is in contrast with what we showed, since we demonstrated lipid fusion induced by UA and OA. One potential explanation might be the fact the presence of each lipid could play a much wider role in membrane structuring in *S. aureus*, as suggested by the formation of neutral pairing between lysylphosphatidylglycerol and phosphatidylglycerol [66]. In the presence of the attracting ion paired phase, the depth of UA or OA penetration into the chains could be altered [66]. The molecular area compared to the unpaired lipids [67] would be expected to also exhibit a reduced headgroup hydration [68]. The reduction in the size of the headgroup hydration shell, compounded by the loss of counter ions, increases the tendency of the lipid ion pairs to adopt a packing geometry with increased negative curvature, thus resulting in the formation of an inverted hexagonal phase. This process could be reinforced by pentacyclic triterpenes as demonstrated for OA in hydrated DPPC system [55] and UA in DPPC and DPPC:Chol model [18]. In the temperature domain of the gel phase, the geometrical form of lipids associated with UA turns into conical form and the self-assembly results in the hexagonal structure. If the ratio of UA increases further, the formation of a cubic structure is also favored. In the temperature domain above the chain melting, there is a significant expansion in the acyl chain cross-sectional area of each lipid molecule, which is increased from 0.4 to 0.7 nm^2^. The actual geometrical form of associated UA and lipids became cylindrical, supporting the bilayer formation. How UA and OA affect lipid polymorphism is probably critical for understanding the mechanism behind the lipid fusion [64,69,70] and the inhibition cytokinesis [66].

In conclusion, UA and OA are able to perturb critical biophysical membrane properties like fluidity and permeability. The disintegration of cardiolipin-enriched domains induced by UA [61] could be linked with PBP2a delocalization from the *S. aureus* septum induced by pentacyclic triterpenes [7]. The activity of the scaffold protein flotillin with reduction of the MRSA proliferative capacity in the presence of β-lactam antibiotics has also been reported in link with perturbation of functional membrane microdomains architecture and efficient PBP2a oligomerization [71,72]. Lastly, other lipid domains like those mostly composed of still-unknown isoprenoid membrane lipids (e.g., staphyloxanthines) [73,74] could be also explored as new targets to address antibiotic resistance [71,75]. Because septum of the bacteria is the peculiar location of PBP, lipoteichoic acids (LTA) and cardiolipin during cell division, it could constitute an opportunity to expand the strategies to design new anti-infective agents, to overcome MRSA antibiotic resistance and to reduce the high mortality rates caused by invasive MRSA infections.

## Figures and Tables

**Figure 1 antibiotics-10-01381-f001:**
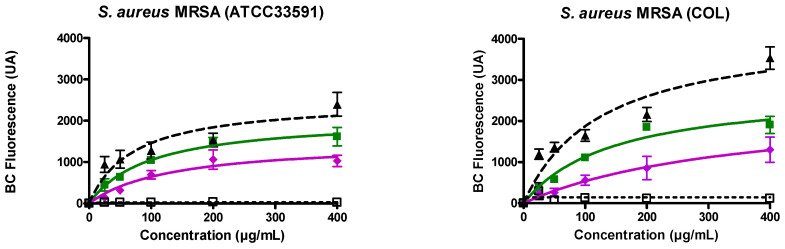
BODIPY^TM^-TR-cadaverine (BC) displacement from its binding to LTA (Lipoteichoic acids) of *S. aureus* (MRSA (ATCC 33591 and COL)) induced by increasing concentrations of alexidine (black triangles, dashed line ▲), UA (Ursolic acid; green squares, ■), OA (Oleanolic acid; pink diamonds, ♦), neamine (black open squares, dotted line □). The data represent the mean ± SEM of three independent experiments.

**Figure 2 antibiotics-10-01381-f002:**
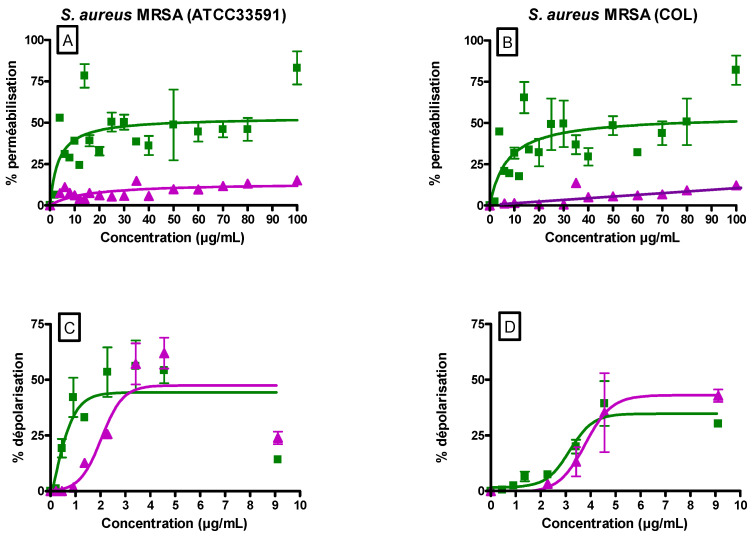
*S. aureus* MRSA (ATCC 33591 et COL) membrane permeabilization (PI fluorescence assay; **A**,**B**) and depolarization (DiSC3(5) fluorescence; **C**,**D**) induced by increasing concentrations of UA (Ursolic acid) (■) and OA (Oleanolic acid) (▲) expressed as a percentage of positive controls considered as 100%: alexidine (5 µM) and valinomycin (10 µM), respectively. The data represent the mean ± SEM of three independent experiments.

**Figure 3 antibiotics-10-01381-f003:**
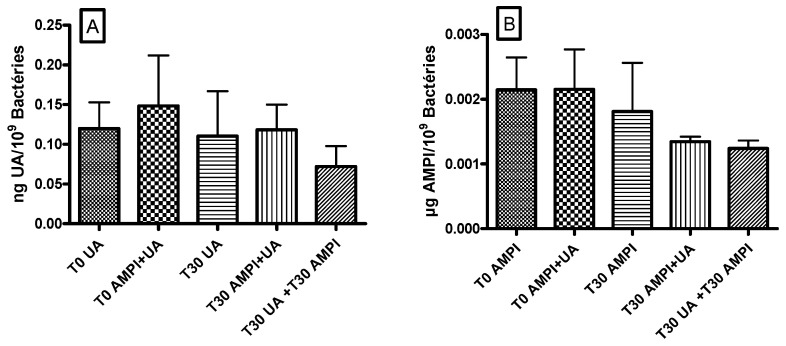
Accumulation of UA (Ursolic acid) (**A**) or AMPI (ampicillin) (**B**) in the presence of ampicillin (**A**) or UA (**B**) in *S. aureus* ATCC 33591. In the co-incubation protocol, both compounds were co-incubated for 30 min (T30 AMPI + UA). For the pre-incubation protocol, UA was incubated for 30 min before the addition of ampicillin for 30 min more (T30 UA + T30 AMPI) or the opposite (ampicillin was incubated for 30 min before the addition of UA for 30 min more (T30 AMPI + T30 AMPI). The accumulation was expressed in ng ursolic acid/10^9^ bacteria or µg ampicillin/10^9^ bacteria. The data represent the mean ± SEM of three independent experiments.

**Figure 4 antibiotics-10-01381-f004:**
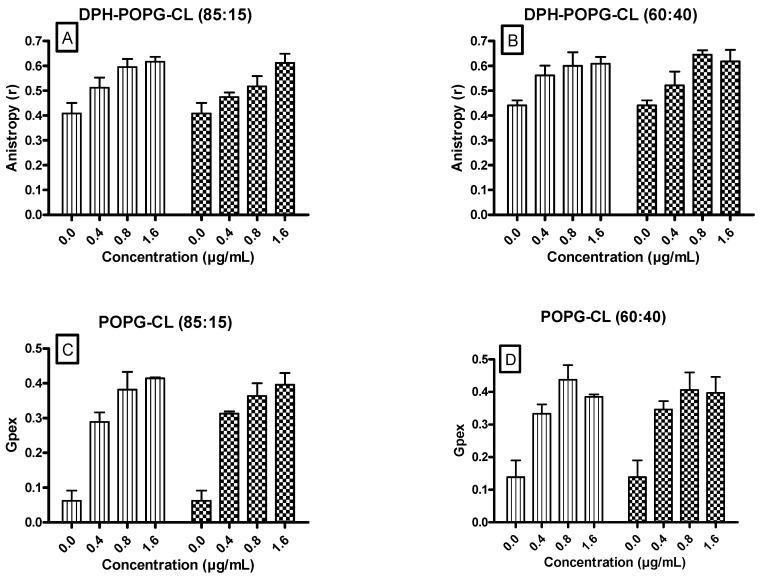
Effect of UA (Ursolic acid) (vertical lines) and OA (Oleanolic acid) (lattice) on fluorescence anisotropy (r) of DPH (Diphenylhexatriene) (**A**,**B**) and on generalized polarization (Gpex) values for Laurdan (**C**,**D**) in lipid vesicles (POPG:CL, 85:15 (**left**) or POPG:CL, 60:40 (**right**)). Data (SEM) are representative of experiments that were reproduced three times. No significant differences were observed when UA and OA were compared. POPG (1-Palmitoyl-2-oleoyl-*sn*-glycero-3-phosphoglycerol), CL (cardiolipin).

**Figure 5 antibiotics-10-01381-f005:**
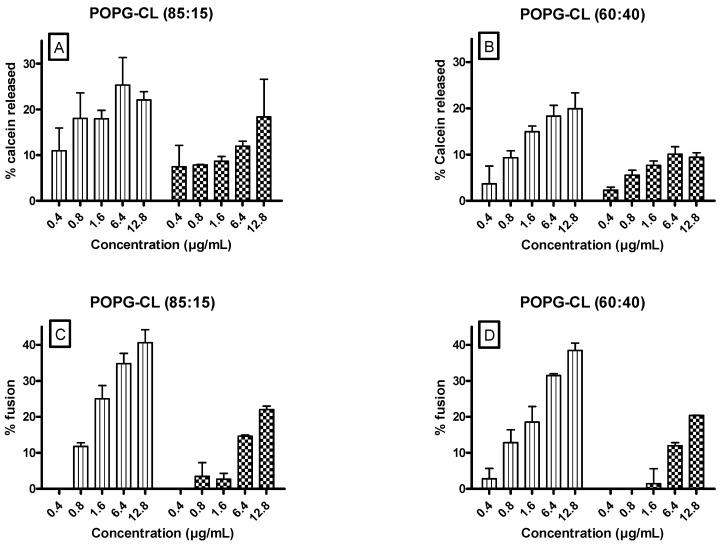
Effect of UA (Ursolic acid) (vertical lines) and OA (Oleanolic acid) (lattice) on calcein release (**A**,**B**) and relief of fluorescence self-quenching of R18s-labeled liposomes (**C**,**D**) in lipid vesicles (POPG:CL, 85:15 (**A**,**C**) or POPG:CL, 60:40 (**B**,**D**). The ordinate shows the percentage of calcein released (**A**,**B**) or the percentage of R18-fluorescence dequenching (**C**,**D**) compared to what was observed after addition of 2% Triton X-100 (positive control, considered as 100%). Data (SEM) are representative of three independent experiments in triplicate. Significant differences were observed when comparing the effect induced by UA and OA. POPG (1-Palmitoyl-2-oleoyl-*sn*-glycero-3-phosphoglycerol), CL (cardiolipin).

**Figure 6 antibiotics-10-01381-f006:**
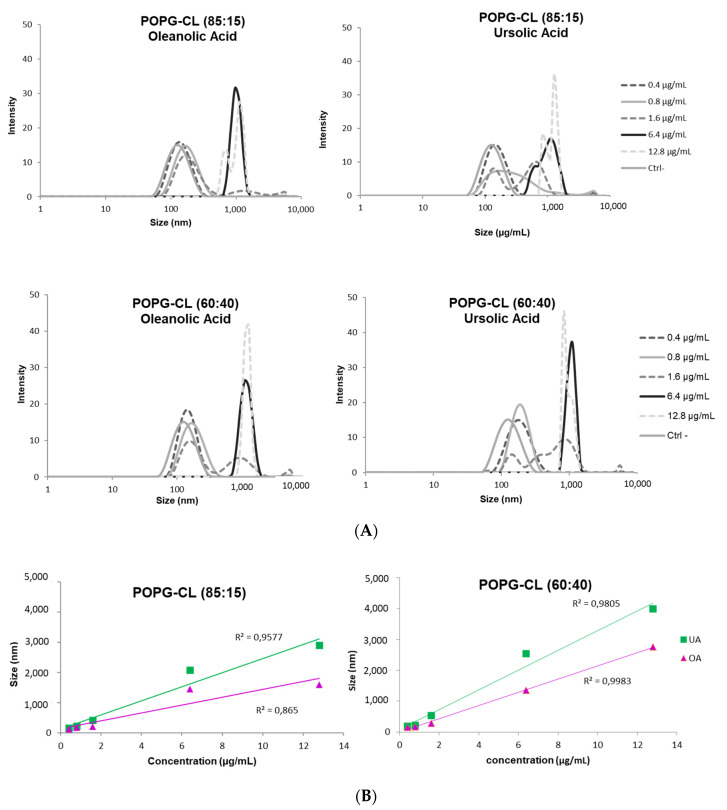
Effect of UA (Ursolic acid) and OA (Oleanolic acid) on LUV size in lipid vesicles (POPG:CL, 85:15 (**top**) or POPG:CL, 60:40 (**bottom**)) (**A**) Effect of UA and OA concentrations on liposome size; (**B**) POPG (1-Palmitoyl-2-oleoyl-*sn*-glycero-3-phosphoglycerol), CL (cardiolipin).

**Table 1 antibiotics-10-01381-t001:** Antimicrobial activity of ursolic acid (UA), oleanolic acid (OA) and ampicillin (AMP) on MRSA strains and FIC index (FICI; Fractional Inhibitory Concentration Indices) of their combinations. N = 3.

*S. aureus*	MICs (mg/L)			FICI	
	UA	OA	AMP	AMP-UA	AMP-OA
ATCC33591	16	32	64	0.38–1.00	0.31–1.00
COL	8	16	32	0.38–1.00	0.25–1.00

## Data Availability

The data presented in this study are available on request from the corresponding author.

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
