# Peer review of "Effect of Ursolic and Oleanolic Acids on Lipid Membranes: Studies on MRSA and Models of Membranes"

_antibiotics, 2021, doi:10.3390/antibiotics10111381_

Round 1

Reviewer 1 Report

Article is well written with scientific grounds on very important topic of alternative to antibiotics. Following comments are presented for improvement in the article

Abstract:

  • What is

Introduction:

Overall, introduction need to be revised. There are several small paragraphs that should be combined to present not more than 4 paragraphs. Write on following grounds about interview

“Importance of MRSA in livestock and public health, its response to antibiotics, need of alternative, UA and OA, its mechanism of action, and objectives”

  • Consider following sentence for grammatical correction

One approach is to benefit from the plant-derived chemicals able to enhance the bacteria susceptibility to antibiotics

  • Following paragraph need to be revised for better understanding

UA and/or OA also showed synergy with other antibiotics………. and colistin-resistant Enterobacteriacae (colistin (Sundaramoorthy et al., 2019)).”

  • What does mean by events in following sentence, Please rephrase it

“These events are likely subsequent to an effect on membrane integrity (Wang et al., 2016; Sundaramoorthy et al., 2019;Qian et al., 2020;Kurek et al., 2010;Han et al., 1997).

  • Second last paragraph “Our work (Catteau et al., 2017) demonstrated reversion of MRSA phenotype by UA….. to design new anti-infective agents (Pinho & Errington, 2005).” Seems not the format of introduction of any article. It seems discussion portion of article. Please correct it.
  • In last paragraph, objectives need to be written in objectively not like subjective form

Materials and Method:

  • Reference CLSI, 2013 is very old, there are new standards available. It is better to confirm your findings according to latest guidelines
  • Are you sure that 106 CFU/mL are required for broth microdilution method? Or its 105 CFU/mL?
  • Catteau et , 2015 has been cited as reference for protocol. Use this reference if they have standardized or have made some modified form of this test, However, it is better if you cite standard protocols.
  • Some of SI units are needed to be written in correct form

Statistical analysis:

  • Please mention statistical tests applied on the data

Results:

  • There are some sentences as repetition of materials and methods e.g “To characterize the combinatorial activity, UA and OA were used in combination with ampicillin at different concentrations”. Please delete such sentences from the results section and write results directly without prior description.
  • There should be description of abbreviation under each table. e.g AMP, OA, MIC, FICI in table 1 and so on.

Discussion

  • Comparison of results from other studies is required. Reasoning is provided in well acceptable manner, however comparison of results with other studies is required

Conclusion:

  • Conclusion needs to be revised to expand finding in this section at least two-three more sentences in addition.

Author Response

Dear reviewer,

We thank you for all very interesting comments you did. Definitively, they help us to improve the paper.

The manuscript has been modified to take into account all comments. The answer to remarks and the revised version are in attached file.

Best regards,

Marie-Paule Mingeot-Leclercq

Reviewer 2 Report

The manuscript presented by Vertraeten et al. describes the overall effects on membranes (both bacterial and lipid models) of two plant-derived antibiotics, ursolic acid (UA) and oleanolic acid (OA). These two compounds have been proposed as potential adjuvants to the antimicrobial treatment of S. Aureus infections with ampicillin, as a way to overcome bacterial drug resistance via a synergistic effect.

The study is particularly relevant as bacterial drug resistance is an ever-growing threat to clinical treatment of infections, and developing novel strategies to bypass such resistance is beneficial for healthcare worldwide.

Overall, the manuscript is well presented and the objectives and methodologies are clearly outlined.  The biophysical methodologies employed on liposomes are in line with the accepted standards in the soft matter community and have been applied properly.

The results are sound and clearly support the interpretation presented by the authors, however, I find some aspects of the study lacking an in-depth interpretation or comments on the mechanistic aspect of the interaction between the compounds and membranes.

I recommend this manuscript for publication provided the authors expand some of the results and discussion on the biophysical investigation, to provide more insight on what the mechanism behind the effect of the membrane biophysical properties might be.

As my expertise mainly lies in lipid membranes and biophysical characterization, I will limit my comments on the second part of the manuscript, where work was conducted on liposomal systems and their properties.

Scientific comments:

  • There is no mention in the materials and methods section on which cardiolipin has been employed for the preparation of LUVs and what the transition temperature for such lipid is. Please provide this information as it would help clarify the LAURDAN GP values presented in the text.
  • There is seems to be little commentary to the results obtained for DPH anisotropy and LAURDAN spectra, and the authors briefly state that UA and OA result in rigidification of the membranes. This is a bit reductive in my opinion, as the results themselves open some mechanistic interpretation of what the interaction of the compounds is. For example:
    • As noted by the authors, changes in DOPG/cardiolipin ratio does not significantly affect the net effect of UA and OA. This in itself could imply that the mechanism might be not dependent on cardiolipin but rather on PG. Did the authors perform a similar experiment on DOPG only or DOPC/cardiolipin LUVs to infer whether the effects are dependent on the charge of the lipids?
    • Linked to the point above, if indeed electrostatic interaction is the main driving force, the physical-chemical properties of UA and OA could be compared to comment on the differences in effect between the two compounds.
    • Although the increase in Laurdan GP is potentially a result of membrane rigidification, it might also mean a difference in miscibility between the two lipid phases. I would recommend performing a temperature scan of the two systems at a fixed UA and OA concentration and compare it to control, to investigate differences in miscibility temperature (this could also be corroborated by DSC if necessary). Also, it is possible to distinguish between single-phase and multi-phase membranes with LAURDAN by comparing GP at 340 nm and 390 nm excitation (see, for example, Morandi et al., PNAS, 2021). Overall, these results could help understand whether the stiffening is an insertion-based effect (like cholesterol) or a clustering effect.
    • In the calcein leakage and R18 membrane mixing experiments (Fig. 5 and Fig. 1S) no negative control (where only buffer and no compound is being added) has been presented, please provide this data as it is necessary to evaluate the net effect of UA and OA.
    • In the results of Fig. 6, the authors comment on the 1000 nm peak appearing upon the addition of UA or OA as a result of the fusion between lipid membranes. Although this may be true, such large-scale objects could also mean vesicles aggregation driven by the compounds. Does the typical size of the LUVs suspension still increase if the size distribution is considered only below 500 nm?

General comments:

  • In the introduction, in the third paragraph to last, it is mentioned “biophysical membranes”. This concept seems a bit puzzling to me. Did the authors perhaps mean the “biophysical properties of membranes”?
  • The materials and methods section is redacted with different fonts, please amend.

Author Response

Dear reviewer,

I would like to thank you for all the very interesting comments and suggestions you did about the manuscript we submitted. We modified the paper accordingly.

The answer to comments and the revised version are in attached file.

Best regards.

Marie-Paule Mingeot

Round 2

Reviewer 2 Report

The revised version provided by the authors has extensively addressed the remarks raised and provided a more in-depth mechanistical interpretation and discussion on the effects of OA and UA on membranes.

I recommend this manuscript for publication.